# A statistical theory of optimal decision-making in sports betting

**Jacek P. Dmochowski** [ID] *

Dept of Biomedical Engineering, City College of New York, New York, NY, United States of America

* jdmochowski@ccny.cuny.edu

## Abstract

The recent legalization of sports wagering in many regions of North America has renewed attention on the practice of sports betting. Although considerable effort has been previously devoted to the analysis of sportsbook odds setting and public betting trends, the principles governing optimal wagering have received less focus. Here the key decisions facing the sports bettor are cast in terms of the probability distribution of the outcome variable and the sportsbook's proposition. Knowledge of the median outcome is shown to be a sufficient condition for optimal prediction in a given match, but additional quantiles are necessary to optimally select the subset of matches to wager on (i.e., those in which one of the outcomes yields a positive expected profit). Upper and lower bounds on wagering accuracy are derived, and the conditions required for statistical estimators to attain the upper bound are provided. To relate the theory to a real-world betting market, an empirical analysis of over 5000 matches from the National Football League is conducted. It is found that the point spreads and totals proposed by sportsbooks capture 86% and 79% of the variability in the median outcome, respectively. The data suggests that, in most cases, a sportsbook bias of only a single point from the true median is sufficient to permit a positive expected profit. Collectively, these findings provide a statistical framework that may be utilized by the betting public to guide decision-making.

## Introduction

The practice of sports betting dates back to the times of Ancient Greece and Rome [1]. With the much more recent legalization of online sports wagering in many regions of North America, the global betting market is projected to reach 140 billion USD by 2028 [2]. Perhaps owing to its ubiquity and market size, sports betting has historically received considerable interest from the scientific community [3].

A topic of obvious relevance to the betting public, and one that has also been the subject of multiple studies, is the efficiency of sports betting markets [4]. While multiple studies have reported evidence for market inefficiencies [5–11], others have reached the opposite conclusion [12, 13]. The discrepancy may signify that certain, but not all, sports markets exhibit inefficiencies. Research into sports betting has also revealed insights into the utility of the "wisdom of the crowd" [14–16], the predictive power of market prices [17–20], quantitative rating

**Competing interests:** The authors have declared that no competing interests exist.

systems [21, 22], and the important finding that sportsbooks exploit public biases to maximize their profits [13, 23].

Indeed, the decisions made by sportsbooks to set the offered odds and payouts have been previously analyzed [13, 23, 24]. On the other hand, arguably less is known about optimality *on the side of the bettor*. The classic paper by Kelly [25] provides the theory for optimizing bet-size (as a function of the likelihood of winning the bet) and can readily be applied to sports wagering. The Kelly bet sizing procedure and two heuristic bet sizing strategies are evaluated in the work of Hvattum and Arntzen [26]. The work of Snowberg and Wolfers [27] provides evidence that the public's exaggerated betting on improbable events may be explained by a model of misperceived probabilities. Wunderlich and Memmert [28] analyze the counterintuitive relationship between the accuracy of a forecasting model and its subsequent profitability, showing that the two are not generally monotonic. Despite these prior works, idealized statistical answers to the critical questions facing the bettor, namely what games to wager on, and on what side to bet, have not been proposed. Similarly, the theoretical limits on wagering accuracy, and under what statistical conditions they may be attained in practice, are unclear.

To that end, the goal of this paper is to provide a statistical framework by which the astute sports bettor may guide their decisions. Wagering is cast in probabilistic terms by modeling the relevant outcome (e.g. margin of victory) as a random variable. Together with the proposed sportsbook odds, the distribution of this random variable is employed to derive a set of propositions that convey the answers to the key questions posed above. This theoretical treatment is complemented with empirical results from the National Football League that instantiate the derived propositions and shed light onto how closely sportsbook prices deviate from their theoretical optima (i.e., those that do not permit positive returns to the bettor).

Importantly, it is *not* an objective of this paper to propose or analyze the utility of any specific predictors ("features") or models. Nevertheless, the paper concludes with an attempt to distill the presented theorems into a set of general guidelines to aid the decision making of the bettor.

## Results

### Problem formulation: "Point spread" betting

A popular form of sports wagering in North American markets is so-called "point spread" betting, where the objective of the bettor is to predict whether the margin of victory will exceed a value proposed by the sportsbook. Here the margin of victory $m \in \mathbb{R}$ is defined as the difference between the number of points obtained by the home team and the number of points obtained by the visiting team:

$$m = \text{home team score} - \text{visiting team score}. \tag{1}$$

Although $m$ is discrete in the vast majority of real-world cases, it is more convenient to work with continuous variables. Throughout, $m$ is modeled as a signed random variable with cumulative distribution function (CDF) $F_m(x) = P(m < x)$.

Next define the *spread $s \in \mathbb{R}$*, which is a proposition set by the sportsbook. In contrast to $m$, the spread is deterministic and known to the bettor. The value of $s$ may be interpreted as the sportsbook's estimate of $m$. In the convention employed here, a value of $s = +3$ denotes that the bookmaker is proposing that the home team will win the match by 3 points. Note that here the spread is not indicated as −3, as is often the case in practice, to emphasize the fact that $s$ is an estimate of $m$.

For positive $s$ (home team favored), the home team is said to "cover the spread" if $m > s$, whereas the visiting team has "beat the spread" otherwise. Conversely, for negative $s$ (visiting

team favored), the visiting team covers the spread if $m < s$, and the home team has beat the spread otherwise. The home (visiting) team is said to win "against the spread" if $m - s$ is positive (negative).

Formally, the objective in point spread betting is to estimate the value of the following Bernoulli random variable:

$$\mathbb{1}_{(s,\infty)}(m) \quad = \quad \begin{cases} 0 & m \notin (s, \infty) \\ 1 & m \in (s, \infty) \end{cases}, \tag{2}$$

where $\mathbb{1}_A(x)$ is the indicator function that takes the value of 1 if $x \in A$ and 0 otherwise.

Denote the profit (on a unit bet) when correctly wagering on the home and visiting teams by $\phi_h$ and $\phi_v$, respectively. Assuming a bet size of $b$ placed on the home team, the conventional payout structure is to award the bettor with $b(1 + \phi_h)$ when $m > s$. The entire wager is lost otherwise. The total profit $\pi$ is thus $b\phi_h$ when correctly wagering on the home team ($-b$ otherwise). When placing a bet of $b$ on the visiting team, the bettor receives $b(1 + \phi_v)$ if $m < s$ and 0 otherwise. Typical values of $\phi_h$ and $\phi_v$ are $100/110 \approx 0.91$, corresponding to a commission of 4.5% charged by the sportsbook.

In practice, the event $m = s$ (termed a "push") may have a non-zero probability and results in all bets being returned. In keeping with the modeling of $m$ by a continuous random variable, here it is assumed that $P(m = s) = 0$. This significantly simplifies the development below. Note also that for fractional spreads (e.g. $s = 3.5$), the probability of a push is indeed zero.

**Wagering to maximize expected profit.** Consider first the question of which team to wager on to maximize the expected profit. As the profit scales linearly with $b$, a unit bet size is assumed without loss of generality.

**Theorem 1** *To maximize the expected profit of a wager, one should bet on the home team if and only if the spread is less than the $\left(\frac{1+\phi_h}{2+\phi_h+\phi_v}\right)$-quantile of $m$.*

*Proof.* Consider the expected profit of the wager, conditioned on the prediction. Assuming that the bettor wagers on the home team, the statistical expectation of profit is:

$$\begin{aligned} E\{\pi | \text{bet home}\} &= P(m > s)\phi_h + P(m \le s)(-1) \\ &= [1 - F_m(s)]\phi_h - F_m(s) \\ &= \phi_h - F_m(s)(1 + \phi_h). \end{aligned} \tag{3}$$

Conversely, a wager on the visiting team has an expected profit of:

$$\begin{aligned} E\{\pi | \text{bet visitor}\} &= P(m \le s)\phi_v + P(m > s)(-1) \\ &= F_m(s)\phi_v - [1 - F_m(s)] \\ &= F_m(s)(\phi_v + 1) - 1. \end{aligned} \tag{4}$$

To maximize the expected profit, the bettor should bet on the home team if and only if:

$$\begin{aligned} F_m(s)(\phi_v + 1) - 1 &< \phi_h - F_m(s)(1 + \phi_h) \\ F_m(s) &< \frac{1 + \phi_h}{2 + \phi_h + \phi_v} \\ F_m(s) &< F_m\left[F_m^{-1}\left(\frac{1 + \phi_h}{2 + \phi_h + \phi_v}\right)\right] \\ s &< F_m^{-1}\left(\frac{1 + \phi_h}{2 + \phi_h + \phi_v}\right), \end{aligned} \tag{5}$$

where the last line follows from the monotonicity of the CDF and where $F_m^{-1}(u) = \inf\{x|F_m(x) \geq u\}$ is the inverse of the CDF of $m$.

**Corollary 1**. *Assuming equal payouts for home and visiting teams ($\phi_h = \phi_v$), maximization of expected profit is achieved by wagering on the home team if and only if the spread is less than the median margin of victory.*

*Proof.* Substituting $\phi_h = \phi_v = \phi$ into (5), one obtains:

$$
\begin{aligned}
s &< F_m^{-1}(1/2) \\
&< \bar{m},
\end{aligned}
\tag{6}
$$

where $\bar{m}$ is the median of $m$.

The significance of (6) is two-fold: picking the side in an optimal way does *not* require knowledge of the distribution of $m$, but rather only its median (or in the general case of (5), a single quantile). Secondly, any estimators of $m$ should be aimed at estimating its median $\bar{m} = F_m^{-1}(1/2)$, and not the mean $\mu_m = E\{m\}$. Note that conventional regression yields estimates of the mean conditioned on some covariates.

A subtle but important point is that knowledge of which side to bet on for each match is insufficient for maximizing overall profit. The reason is that even if wagering on the side with higher expected profit, it is possible (and in fact quite common, see empirical results below) that the "optimal" wager carries a negative expectation. Thus, an understanding of when wagering should be avoided altogether is required. This is the subject of the theorem below.

**Theorem 2**. *A positive expected profit is only possible if the spread is less than the $\left(\frac{\phi_h}{1+\phi_h}\right)$-quantile, or greater than the $\left(\frac{1}{1+\phi_v}\right)$-quantile of m.*

*Proof.* This follows from the expected profit conditioned on the side. From (3), a wager on the home team carries a positive expectation when:

$$
\phi_h - F_m(s)(1 + \phi_h) > 0 \quad,
$$

leading to:

$$
s < F_m^{-1}\left(\frac{\phi_h}{1+\phi_h}\right).
$$

Conversely, from (4), a wager on the visiting team has a positive profit when:

$$
\begin{aligned}
F_m(s)(\phi_v + 1) - 1 &> 0 \\
s &> F_m^{-1}\left(\frac{1}{1+\phi_v}\right).
\end{aligned}
$$

It is instructive to consider the conditions above for typical values of $\phi_h$ and $\phi_v$. When wagering on the home team with $\phi_h = 0.91$, positive expectation requires the spread to be no larger than the 0.476 quantile of $m$. When wagering on the visiting team, the spread must exceed the 0.524 quantile. This means that, *if the spread is contained within the 0.476-0.524 quantiles of the margin of victory, wagering should be avoided*. Practically, it is thus important to obtain estimates of this interval and its proximity to the median score in *units of points*.

The result of Theorem 2 is reminiscent of the "area of no profitable bet" scenario described in [28]. Whereas the latter result is presented in terms of outcome probabilities estimated by the bettor and the sportsbook, Theorem 2 here delineates the conditions under which the sportsbook's point spread assures a negative expectation on the bettor's side.

**Optimal estimation of the margin of victory.** In practice, the margin of victory must be estimated from available data. Denote the estimate of the margin by $\hat{m}$, a random variable with

a sampling distribution given by $\hat{F}_m(x) = P(\hat{m} < x)$. Note that the randomness in $\hat{m}$ stems from the sample of data used to compute $\hat{m}$, whereas the randomness in $m$ originates from factors that affect the outcome of the match, such as weather and variable player performance. Given that these are temporally non-overlapping sources of variability—the sources of noise affecting $\hat{m}$ exert influence on the resulting estimate before the sources of noise have begun to exert their influence on the outcome of the match—it is assumed that, for a given match, $m$ and $\hat{m}$ are independent:

$$P(m, \hat{m} \mid \theta) = P(m|\theta)P(\hat{m}|\theta), \tag{7}$$

where $\theta$ captures the identity of the two teams and all other factors that define a particular match. Below the dependence on $\theta$ is omitted for notational convenience.

**Theorem 3**. *Define an "error" as a wager that is placed on the team that loses against the spread. The probability of error is bounded according to*: $\min\{F_m(s), 1 - F_m(s)\} \leq p(\text{error}) \leq \max\{F_m(s), 1 - F_m(s)\}$.

*Proof.* Such an error is made when $\hat{m}$ and $m$ fall on opposite sides of the spread $s$. From the axioms of probability, this event has a probability of:

$$\begin{aligned} p(\text{error}) &= P(\hat{m} \leq s)P(m > s) + P(\hat{m} > s)P(m \leq s) \\ &= \hat{F}_m(s)[1 - F_m(s)] + [1 - \hat{F}_m(s)]F_m(s) \\ &= F_m(s) + \hat{F}_m(s)[1 - 2F_m(s)]. \end{aligned} \tag{8}$$

Optimization of $p(\text{error})$ with respect to $\hat{F}_m(s)$ is a linear programming problem. To derive the upper bound, consider the following optimization:

$$\max_{\hat{F}_m(s)} p(\text{error}) \quad \text{subject to } 0 \leq \hat{F}_m(s) \leq 1. \tag{9}$$

When $1 - 2F_m(s) > 0$, $p(\text{error})$ is clearly maximized when $\hat{F}_m(s) = 1$, where it attains a maximum value of $1 - F_m(s)$. On the other hand, when $1 - 2F_m(s) < 0$, $p(\text{error})$ is maximized when $\hat{F}_m(s) = 0$, when it attains a value of $F_m(s)$. By the same reasoning, the minimum value of $p$(error) is $F_m(s)$ when $1 - 2F_m(s) > 0$, and $1 - F_m(s)$ when $1 - 2F_m(s) < 0$. Putting this all together, one obtains the required bounds.

The result of Theorem (8) provides both the best- and worst-case scenario of a given wager. When $F_m(s)$ is close to 1/2, both the minimum and maximum error rates are near 50%, and wagering is reduced to an event akin to a coin flip. On the other hand, when the true median is far from the spread (i.e., $F_m(s)$ deviates from 1/2), the minimum and maximum error rates diverge, increasing the highest achievable accuracy of the wager.

**Theorem 4**. *Define an "excess error" as a wager that is placed on the team that does not maximize expected profit. Any estimator that satisfies $\hat{F}_m(s) = \mathbb{1}_{(-\infty,s)}(\bar{m})$ minimizes the probability of excess error.*

*Proof.* By definition, the excess error is given by:

$$p(\text{excess error}) = p(\text{error}) - \min\{F_m(s), 1 - F_m(s)\}. \tag{10}$$

When $F_m(s) \leq 1 - F_m(s)$, the excess error follows from (8) as:

$$\begin{aligned} p(\text{excess error}) &= p(\text{error}) - F_m(s) \\ &= \hat{F}_m(s)[1 - 2F_m(s)]. \end{aligned} \tag{11}$$

It then follows that the excess error is minimized by an estimator whose CDF evaluates to 0 at the spread: $\hat{F}_m(s) = 0$. Similarly, when $F_m(s) > 1 - F_m(s)$, the excess error is written as:

$$
\begin{aligned}
p(\text{excess error}) &= p(\text{error}) - [1 - F_m(s)] \\
&= \left[1 - \hat{F}_m(s)\right][2F_m(s) - 1],
\end{aligned}
\tag{12}
$$

which is minimized by any estimator satisfying $\hat{F}_m(s) = 1$. Noting that $F_m(s) \leq 1 - F_m(s)$ is equivalent to $F_m(s) \leq 1/2$, it follows that:

$$
\begin{aligned}
\hat{F}_m(s) &= \begin{cases} 0 & F_m(s) \leq 1/2 \\ 1 & F_m(s) > 1/2 \end{cases} \\
&= \mathbb{1}_{(-\infty, s)}(\bar{m}).
\end{aligned}
$$

The significance of this result is that an optimal estimator of $m$ need not be close to the true median $\bar{m}$. Rather, the estimator degrees of freedom should aim to generate predictions $\hat{m}$ that are on the same side of $s$ as the true value. In statistical terms, an optimal estimator may possess a large bias.

## Optimality in "moneyline" wagering

A popular type of sports wager is the so-called "moneyline" bet, where the task of the bettor is to predict which side will win the match, regardless of the magnitude of the margin of victory. Mathematically, the objective of this wager is to predict the sign of $m$, which is a special case of point spread betting where $s = 0$. The primary difference between point spread and moneyline wagering is expressed in the magnitudes of $\phi_h$ and $\phi_v$. Whereas point spread betting has $\phi_h/\phi_v \approx 1$, the ratio of home to visitor payouts exhibit a larger dynamic range in moneyline wagering:

$$
\frac{1}{K} \leq \frac{\phi_h}{\phi_v} \leq K,
\tag{13}
$$

where $K$ is a large positive number. The deviation of $\frac{\phi_h}{\phi_v}$ from 1 reflects the perceived imbalance in the quality of the two sides. When the home team is strongly favored to win, $\frac{\phi_h}{\phi_v}$ is close to 0, whereas $\frac{\phi_h}{\phi_v}$ is large when the visiting team is heavily favored. The following results follow from substituting $s = 0$ into Theorems 1 to 4.

**Corollary 2**. *To maximize the expected profit of a moneyline wager, one should bet on the home team if and only if the $\left(\frac{1+\phi_h}{2+\phi_h+\phi_v}\right)$-quantile of m is positive.*

**Corollary 3**. *In moneyline wagering, a positive expected profit is only possible if the the $\left(\frac{\phi_h}{1+\phi_h}\right)$-quantile of m is positive, or if the $\left(\frac{1}{1+\phi_v}\right)$-quantile of m is negative.*

**Corollary 4**. *Define an "error" as a wager that is placed on the team that loses the match outright. The probability of error in moneyline wagering is bounded according to*: $\min\{F_m(0), 1 - F_m(0)\} \leq p(\text{error}) \leq \max\{F_m(0), 1 - F_m(0)\}$.

**Corollary 5**. *Define an "excess error" as a wager that is placed on the side that does not maximize the expected profit of a moneyline wager. Any estimator that satisfies $\hat{F}_m(0) = \mathbb{1}_{(-\infty, 0)}(\bar{m})$ minimizes the probability of an excess error.*

Notice that optimal decision-making in moneyline wagers requires knowledge of quantiles that may be near 0 (if $\phi_v \gg \phi_h$) or near 1 (if $\phi_h \gg \phi_v$). More subtly, the required quantiles will

differ for matches that exhibit different payout ratios. For example, a match with two even sides will require knowledge of central quantiles, while a match with a 4:1 favorite will require knowledge of the 80th and 20th percentiles. The implications of this property on quantitative modeling are described in the *Discussion*.

The moneyline wagering considered in this section is a two-alternative bet that is popular in North American sports. In European betting markets, the most common type of wager is the three-alternative "Home-Draw-Away" bet where there is no point spread and the task of the bettor is to forecast one of the three potential outcomes: $m > 0$, $m = 0$, or $m < 0$, each of which are endowed with a payout (see, for example, [26, 29, 30]). Clearly the the probability $p(m = 0)$ will be non-zero in this context. As a result, the methodology here, which models $m$ by a continuous random variable, cannot be straightforwardly applied to the case of the Home-Draw-Away bet. The extension of the present findings to the case of multi-way bets with discrete $m$ is a potential topic of future research.

## Optimality in "over-under" betting

In "over-under" or "total" wagering, the objective of the bettor is to predict whether the total number of points obtained by both sides:

$$t = \text{home team score} + \text{visiting team score} \tag{14}$$

exceeds a proposition $\tau$, where $\tau$ may be viewed as the sportsbook's estimate of $t$. When correctly predicting that $t > \tau$ ("over"), the bettor is awarded with a profit $\pi = b\phi_o$. Similarly, when correctly predicting that $t < \tau$ ("under"), the bettor receives a profit of $\pi = b\phi_u$. The entire wager is lost when the prediction is incorrect. In the event $\tau = t$, all bets are returned. It is thus clear that over-under betting is mathematically equivalent to point spread wagering, with the margin of victory $m$ replaced by $t$ as the target variable. Analogous to point-spread betting, typical values for $\phi_o$ and $\phi_u$ are 0.91.

The following two results may be proven by replacing $m$ with $\tau$, $\phi_h$ with $\phi_o$, and $\phi_v$ with $\phi_u$ in the Proofs of Theorems 1 and 2, respectively.

**Corollary 6**. *To maximize the expected profit of an over-under wager, one should wager on the "over" ($t > \tau$) if and only if $\tau$ is less than the $\left(\frac{1+\phi_o}{2+\phi_o+\phi_u}\right)$-quantile of $t$.*

In the special case of $\phi_o = \phi_u$, one should bet on the over only if and only if the sportsbook total $\tau$ falls below the median of $t$.

**Corollary 7**. *In over-under betting, a positive expected profit is only possible if the sportsbook total $\tau$ is less than the $\left(\frac{\phi_o}{1+\phi_o}\right)$-quantile, or greater than the $\left(\frac{1}{1+\phi_u}\right)$-quantile, of $t$.*

Define $F_t(\tau)$ as the CDF of the true point total evaluated at the sportsbook's proposed total. The following corollary may be proven by following the Proof of Theorem 3.

**Corollary 8**. *Define an "error" in over-under betting as a wager that is placed on the "over" when $t < \tau$ or on the "under" when $t > \tau$. The probability of error is bounded according to*: $\min\{F_t(\tau), 1 - F_t(\tau)\} \leq p(\text{error}) \leq \max\{F_t(\tau), 1 - F_t(\tau)\}$.

Define $\hat{t}$ as the bettor's estimate of $t$, and $\hat{F}_t$ as the CDF of the sampling distribution of $\hat{t}$. The following result may be proven by replacing $\hat{F}_m(s)$ with $\hat{F}_t(\tau)$ in the Proof of Theorem 4.

**Corollary 9**. *Define an "excess error" as a wager that is placed on the outcome (over or under) that does not maximize expected profit. Any estimator that satisfies $\hat{F}_t(\tau) = \mathbb{1}_{(-\infty,\tau)}(\bar{t})$ minimizes the probability of excess error.*

## Empirical results from the National Football League

In order to connect the theory to a real-world betting market, empirical analyses utilizing historical data from the National Football League (NFL) were conducted. The margins of victory, point totals, sportsbook point spreads, and sportsbook point totals were obtained for all regular season matches occurring between the 2002 and 2022 seasons ($n = 5412$). The mean margin of victory was $2.19 \pm 14.68$, while the mean point spread was $2.21 \pm 5.97$. The mean point total was $44.43 \pm 14.13$, while the mean sportsbook total was $43.80 \pm 4.80$. The standard deviation of the margin of victory is nearly 7x the mean, indicating a high level of dispersion in the margin of victory, perhaps due to the presence of outliers. Note that the standard deviation of a random variable provides an upper bound on the distance between its mean and median [31], which is relevant to the problem at hand.

To estimate the distribution of the margin of victory for individual matches, the point spread $s$ was employed as a surrogate for $\theta$. The underlying assumption is that matches with an identical point spread exhibit margins of victory drawn from the same distribution. Observations were stratified into 21 groups ranging from $s_o = -7$ to $s_o = 10$. This procedure was repeated for the analysis of point totals, where observations were stratified into 24 groups ranging from $t_o = 37$ to $t_o = 49$.

**How accurately do sportsbooks capture the median outcome?.**   It is important to gain insight into how accurately the point spreads proposed by sportsbooks capture the median margin of victory. For each stratified sample of matches, the median margin of victory was computed and compared to the sample's point spread. The distribution of margin of victory for matches with a point spread $s_o = 6$ is shown in Fig 1a, where the sample median of 4.34 (95% confidence interval [2.41,6.33]; median computed with kernel density estimation to overcome the discreteness of the margin of victory; confidence interval computed with the bootstrap) is lower than the sportsbook point spread. However, the sportsbook value is contained within the 95% confidence interval.

Aggregating across stratified samples, the sportsbook point spread explained 86% of the variability in the true median margin of victory ($r^2 = 0.86$, $n = 21$; Fig 1c). Both the slope (0.93, 95% confidence interval [0.81,1.04]) and intercept (-0.41, 95% confidence interval [-1.03,0.16]) of the ordinary least squares (OLS) line of best fit (dashed blue line) indicate a slight overestimation of the margin of victory by the point spread. This is most apparent for positive spreads (i.e., a home favorite). Nevertheless, the confidence intervals of both the slope and intercept did include the null hypothesis values of 1 and 0, respectively. The data for all sportsbook point spreads with at least 100 matches is provided in Table 1.

The distribution of observed point totals for matches with a sportsbook total of $\tau = 46$ is shown in Fig 1b, where the computed median of 44.45 (95% confidence interval [42.25,46.81]) is suggestive of a slight overestimation of the true total. Combining data from all samples, the sportsbook point total explained 79% of the variability in the median point total ($r^2 = 0.79$, $n = 24$; Fig 1d).

Interestingly, the data hints at the sportsbook's proposed point total *underestimating* the true total for relatively low totals (i.e., black line is below the blue for sportsbook totals below 43), while overestimating the total for those matches expected to exhibit high scoring (i.e., black line is above the blue line for sportsbook totals above 43). Note, however, that the confidence intervals of the regression line (slope: [0.72,1.02], intercept: [-1.14, 12.05]) did contain the null hypothesis values. The data for all sportsbook point total with at least 100 samples is provided in Table 2.

**Do sportsbook estimates deviate from the 0.476-0.524 interval?.**   In the common case of $\phi = 0.91$, a positive expected profit is only feasible if the point spread (or point total) is either

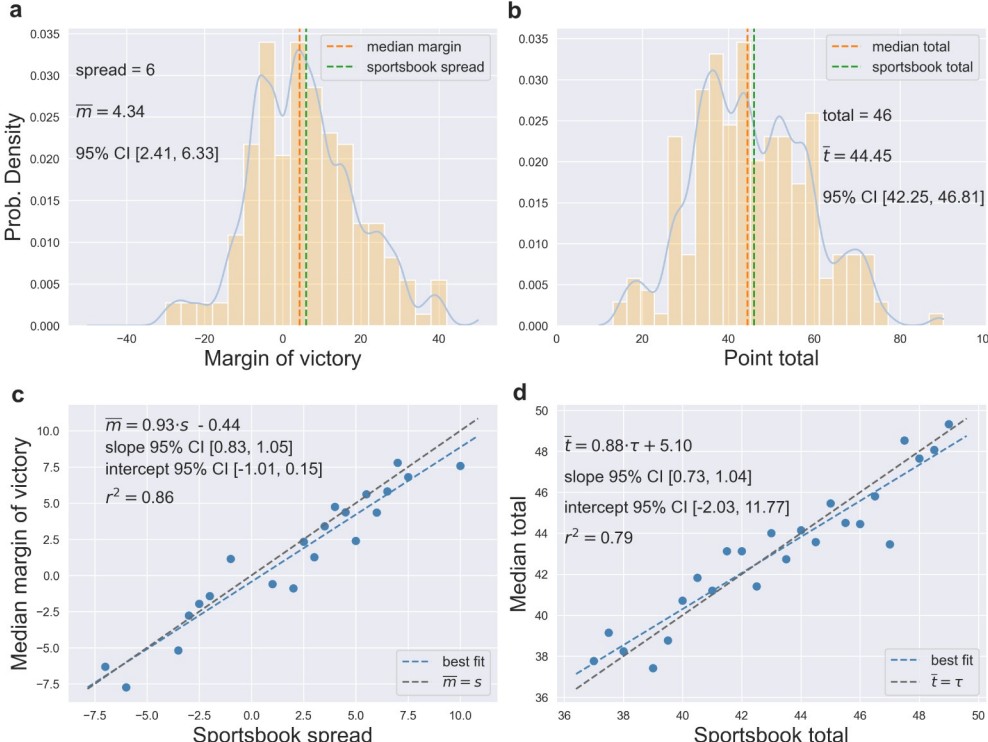

**Fig 1. How accurately do sportsbooks predict the median outcome?** (**a**) The distribution of margin of victory for National Football League matches with a consensus sportsbook point spread of $s = 6$. The median outcome of 4.26 (dashed orange line, computed with kernel density estimation) fell below the sportsbook point spread (dashed blue line). However, the 95% confidence interval of the sample median (2.27-6.38) contained the sportsbook proposition of 6. (**b**) Same as (a), but now showing the distribution of point total for all matches with a sportsbook point total of 46. Although the sportsbook total exceeded the median outcome by approximately 1.5 points, the confidence interval of the sample median (42.25-46.81) contained the sportsbook's proposition. (**c**) Combining all stratified samples, the sportsbook's point spread explained 86% of the variability in the median margin of victory. The confidence intervals of the regression line's slope and intercept included their respective null hypothesis values of 1 and 0, respectively. (**d**) The sportsbook point total explained 79% of the variability in the median total. Although the data hints at an overestimation of high totals and underestimation of low totals, the confidence intervals of the slope and intercept contained the null hypothesis values.

below the 0.476 or above the 0.524 quantiles of the outcome's distribution. It is thus interesting to consider how often this may occur in a large betting market such as the NFL. To that end, the 0.476 and 0.524 quantiles of the margin of victory were estimated in each stratified sample (horizontal bars in Fig 2; the point spread is indicated with an orange marker; all quantiles are listed in Table 1).

For the majority of samples, the confidence intervals of the 0.476 and 0.524 quantiles contained the sportsbook spread. One exception was the spread $s = 5$, where the margin of victory fell below the sportsbook value (95% confidence interval of the 0.524 quantile: [0.87,4.85]). The margin of victory for $s = 3$ (95% confidence interval of the 0.524 quantile: [0.78,3.08]) and $s = 10$ (95% confidence interval of the 0.524 quantile: [6.42,10.06]) also tended to underestimate the sportsbook spread, with the confidence intervals just containing the sportsbook value.

The analysis was repeated for point totals (Fig 3, all quantiles listed in Table 2). All but one stratified sample exhibited 0.476 and 0.524 quantiles whose confidence intervals contained the sportsbook total ($t = 47$, [41.59, 45.42]). Examination of the sample quantiles suggests that

**Table 1. The relationship between sportsbook point spread and true margin of victory.** Regular season matches from the National Football League occurring between 2002-2022 were stratified according to their sportsbook point spread. Each set of 3 grouped rows corresponds to a subsample of matches with a common sportsbook point spread. The "level" column indicates whether the row pertains to the 95% confidence interval (0.025 and 0.975 quantiles) or the mean value across bootstrap resamples. The dependent variables include the 0.476, 0.5, and 0.524 quantiles, as well as the expected profit of wagering on the side with higher likelihood of winning the bet for hypothetical point spreads that deviate from the median outcome by 1, 2, and 3 points, respectively.

| Spread | Level | 0.476 | Median | 0.524 | $E\{\pi \mid s = \bar{m} + k\}$ | | | | | | |
|---|---|---|---|---|---|---|---|---|---|---|---|
| | | | | | $k = -3$ | $k = -2$ | $k = -1$ | $k = 0$ | $k = 1$ | $k = 2$ | $k = 3$ |
| -7.0 | 0.025 | -10.433 | -9.473 | -8.453 | 0.056 | 0.024 | -0.011 | -0.045 | -0.008 | 0.032 | 0.079 |
| | 0.975 | -4.511 | -3.891 | -3.390 | 0.183 | 0.117 | 0.041 | -0.045 | 0.050 | 0.150 | 0.245 |
| | Mean | -7.123 | -6.303 | -5.547 | 0.112 | 0.065 | 0.012 | -0.045 | 0.019 | 0.088 | 0.161 |
| -6.0 | 0.025 | -12.214 | -11.314 | -10.453 | 0.056 | 0.023 | -0.011 | -0.045 | -0.009 | 0.028 | 0.066 |
| | 0.975 | -5.591 | -5.031 | -4.511 | 0.176 | 0.114 | 0.041 | -0.045 | 0.056 | 0.167 | 0.279 |
| | Mean | -8.577 | -7.726 | -6.933 | 0.111 | 0.062 | 0.010 | -0.045 | 0.017 | 0.086 | 0.163 |
| -3.5 | 0.025 | -7.533 | -6.932 | -6.332 | 0.136 | 0.076 | 0.015 | -0.045 | 0.013 | 0.065 | 0.113 |
| | 0.975 | -4.090 | -3.431 | -2.771 | 0.237 | 0.147 | 0.052 | -0.045 | 0.051 | 0.143 | 0.226 |
| | Mean | -5.757 | -5.173 | -4.582 | 0.185 | 0.111 | 0.034 | -0.045 | 0.032 | 0.106 | 0.171 |
| -3.0 | 0.025 | -4.452 | -3.931 | -3.331 | 0.151 | 0.082 | 0.015 | -0.045 | 0.008 | 0.056 | 0.103 |
| | 0.975 | -2.270 | -1.550 | -0.730 | 0.223 | 0.137 | 0.046 | -0.045 | 0.039 | 0.111 | 0.172 |
| | Mean | -3.387 | -2.777 | -2.120 | 0.186 | 0.110 | 0.031 | -0.045 | 0.023 | 0.082 | 0.135 |
| -2.5 | 0.025 | -4.191 | -3.691 | -3.172 | 0.119 | 0.059 | 0.004 | -0.045 | -0.001 | 0.042 | 0.085 |
| | 0.975 | -0.690 | 0.111 | 1.071 | 0.256 | 0.164 | 0.060 | -0.045 | 0.056 | 0.144 | 0.211 |
| | Mean | -2.572 | -1.969 | -1.314 | 0.192 | 0.116 | 0.034 | -0.045 | 0.026 | 0.087 | 0.141 |
| -2.0 | 0.025 | -5.612 | -4.892 | -4.091 | 0.051 | 0.018 | -0.014 | -0.045 | -0.013 | 0.019 | 0.055 |
| | 0.975 | 1.310 | 1.991 | 2.651 | 0.179 | 0.104 | 0.030 | -0.045 | 0.036 | 0.123 | 0.208 |
| | Mean | -2.335 | -1.425 | -0.521 | 0.113 | 0.060 | 0.007 | -0.045 | 0.008 | 0.063 | 0.121 |
| -1.0 | 0.025 | -2.371 | -1.611 | -0.890 | 0.071 | 0.033 | -0.007 | -0.045 | -0.006 | 0.034 | 0.080 |
| | 0.975 | 2.732 | 3.311 | 3.831 | 0.186 | 0.113 | 0.037 | -0.045 | 0.049 | 0.151 | 0.252 |
| | Mean | 0.313 | 1.151 | 1.939 | 0.121 | 0.064 | 0.010 | -0.045 | 0.016 | 0.086 | 0.161 |
| 1.0 | 0.025 | -5.412 | -4.432 | -3.411 | 0.025 | -0.001 | -0.024 | -0.045 | -0.024 | -0.000 | 0.027 |
| | 0.975 | 2.071 | 2.771 | 3.511 | 0.154 | 0.084 | 0.020 | -0.045 | 0.028 | 0.109 | 0.187 |
| | Mean | -1.814 | -0.589 | 0.580 | 0.076 | 0.034 | -0.005 | -0.045 | -0.003 | 0.044 | 0.096 |
| 2.0 | 0.025 | -3.931 | -3.371 | -2.811 | 0.082 | 0.033 | -0.008 | -0.045 | -0.010 | 0.027 | 0.064 |
| | 0.975 | 1.431 | 2.371 | 3.171 | 0.256 | 0.157 | 0.054 | -0.045 | 0.043 | 0.118 | 0.177 |
| | Mean | -1.651 | -0.901 | -0.091 | 0.168 | 0.092 | 0.020 | -0.045 | 0.013 | 0.066 | 0.115 |
| 2.5 | 0.025 | -0.471 | 0.410 | 1.250 | 0.096 | 0.049 | 0.002 | -0.045 | 0.006 | 0.063 | 0.124 |
| | 0.975 | 3.372 | 3.971 | 4.511 | 0.183 | 0.118 | 0.042 | -0.045 | 0.051 | 0.151 | 0.250 |
| | Mean | 1.640 | 2.330 | 2.972 | 0.136 | 0.081 | 0.021 | -0.045 | 0.028 | 0.108 | 0.191 |
| 3.0 | 0.025 | -0.810 | 0.010 | 0.830 | 0.105 | 0.054 | 0.004 | -0.045 | 0.008 | 0.067 | 0.130 |
| | 0.975 | 1.770 | 2.431 | 3.051 | 0.158 | 0.092 | 0.027 | -0.045 | 0.035 | 0.119 | 0.200 |
| | Mean | 0.511 | 1.264 | 1.962 | 0.131 | 0.072 | 0.015 | -0.045 | 0.021 | 0.094 | 0.166 |
| 3.5 | 0.025 | 1.090 | 1.830 | 2.511 | 0.105 | 0.059 | 0.010 | -0.045 | 0.015 | 0.077 | 0.139 |
| | 0.975 | 4.311 | 4.872 | 5.471 | 0.200 | 0.129 | 0.046 | -0.045 | 0.048 | 0.139 | 0.224 |
| | Mean | 2.779 | 3.394 | 3.988 | 0.153 | 0.095 | 0.029 | -0.045 | 0.032 | 0.109 | 0.183 |
| 4.0 | 0.025 | 2.651 | 3.271 | 3.811 | 0.125 | 0.076 | 0.018 | -0.045 | 0.022 | 0.089 | 0.157 |
| | 0.975 | 5.711 | 6.232 | 6.752 | 0.245 | 0.160 | 0.060 | -0.045 | 0.062 | 0.165 | 0.261 |
| | Mean | 4.208 | 4.746 | 5.276 | 0.185 | 0.117 | 0.039 | -0.045 | 0.042 | 0.127 | 0.208 |
| 4.5 | 0.025 | 0.610 | 1.510 | 2.371 | 0.077 | 0.037 | -0.004 | -0.045 | -0.004 | 0.038 | 0.080 |
| | 0.975 | 6.432 | 7.252 | 8.032 | 0.193 | 0.117 | 0.037 | -0.045 | 0.038 | 0.120 | 0.201 |
| | Mean | 3.610 | 4.376 | 5.133 | 0.131 | 0.075 | 0.016 | -0.045 | 0.017 | 0.078 | 0.139 |

*(Continued)*

**Table 1.** (Continued)

| Spread | Level | 0.476 | Median | 0.524 | $E\{\pi|s = \bar{m} + k\}$ | | | | | | |
|---|---|---|---|---|---|---|---|---|---|---|---|
| | | | | | $k = -3$ | $k = -2$ | $k = -1$ | $k = 0$ | $k = 1$ | $k = 2$ | $k = 3$ |
| 5.0 | **0.025** | -0.551 | 0.170 | 0.890 | 0.104 | 0.056 | 0.006 | -0.045 | 0.008 | 0.062 | 0.119 |
| | **0.975** | 3.851 | 4.451 | 5.032 | 0.226 | 0.146 | 0.055 | -0.045 | 0.059 | 0.160 | 0.251 |
| | **Mean** | 1.770 | 2.390 | 2.994 | 0.163 | 0.098 | 0.029 | -0.045 | 0.032 | 0.109 | 0.182 |
| 5.5 | **0.025** | 2.910 | 3.610 | 4.191 | 0.117 | 0.065 | 0.011 | -0.045 | 0.010 | 0.063 | 0.112 |
| | **0.975** | 7.032 | 7.672 | 8.333 | 0.238 | 0.148 | 0.052 | -0.045 | 0.050 | 0.141 | 0.225 |
| | **Mean** | 5.004 | 5.613 | 6.228 | 0.176 | 0.106 | 0.031 | -0.045 | 0.030 | 0.101 | 0.168 |
| 6.0 | **0.025** | 1.530 | 2.410 | 3.150 | 0.089 | 0.045 | 0.001 | -0.045 | 0.002 | 0.049 | 0.095 |
| | **0.975** | 5.671 | 6.333 | 7.172 | 0.187 | 0.117 | 0.037 | -0.045 | 0.036 | 0.114 | 0.187 |
| | **Mean** | 3.618 | 4.343 | 5.067 | 0.136 | 0.080 | 0.018 | -0.045 | 0.018 | 0.081 | 0.141 |
| 6.5 | **0.025** | 3.911 | 4.391 | 4.871 | 0.153 | 0.085 | 0.018 | -0.045 | 0.013 | 0.064 | 0.113 |
| | **0.975** | 6.832 | 7.432 | 8.094 | 0.263 | 0.165 | 0.060 | -0.045 | 0.054 | 0.144 | 0.223 |
| | **Mean** | 5.264 | 5.817 | 6.395 | 0.208 | 0.126 | 0.039 | -0.045 | 0.033 | 0.104 | 0.166 |
| 7.0 | **0.025** | 5.111 | 5.891 | 6.711 | 0.090 | 0.042 | -0.003 | -0.045 | -0.006 | 0.032 | 0.069 |
| | **0.975** | 8.813 | 9.733 | 10.734 | 0.173 | 0.101 | 0.027 | -0.045 | 0.024 | 0.086 | 0.141 |
| | **Mean** | 6.973 | 7.783 | 8.645 | 0.133 | 0.072 | 0.012 | -0.045 | 0.008 | 0.057 | 0.103 |
| 7.5 | **0.025** | 4.111 | 4.631 | 5.251 | 0.113 | 0.056 | 0.004 | -0.045 | 0.002 | 0.049 | 0.094 |
| | **0.975** | 8.432 | 9.192 | 9.912 | 0.260 | 0.159 | 0.056 | -0.045 | 0.048 | 0.137 | 0.218 |
| | **Mean** | 6.161 | 6.799 | 7.461 | 0.186 | 0.107 | 0.029 | -0.045 | 0.025 | 0.091 | 0.154 |
| 10.0 | **0.025** | 5.370 | 5.911 | 6.392 | 0.158 | 0.088 | 0.022 | -0.045 | 0.020 | 0.078 | 0.133 |
| | **0.975** | 8.653 | 9.173 | 9.813 | 0.277 | 0.180 | 0.072 | -0.044 | 0.069 | 0.175 | 0.267 |
| | **Mean** | 7.066 | 7.570 | 8.081 | 0.216 | 0.134 | 0.046 | -0.045 | 0.044 | 0.127 | 0.199 |

NFL sportsbooks are very adept at proposing point totals that fall within 2.4 percentiles of the median outcome.

**How large of a discrepancy from the median is required for profit?.** In practice, it is desirable to have an understanding of how large of a sportsbook bias, in units of points, is required to permit a positive expected profit. To address this, the value of the empirically measured CDF of the margin of victory was evaluated at offsets of 1, 2, and 3 points from the true median in each direction. The resulting value was then converted into the expected value of profit (see *Materials and Methods*). The computation was performed separately within each stratified sample, and the height of each bar in Fig 4 indicates the hypothetical expected profit of a unit bet *when wagering on the team with the higher probability of winning against the spread*. For the sake of clarity, only the four largest samples ($s \in \{-3, 2.5, 3, 7\}$) are shown in the Figure, with data for all samples listed in Table 1.

The expected profit is negative (i.e., $(\phi - 1)/2 = -0.045$) when the spread equals the median (center column). Interestingly however, for 3 of the 4 largest stratified samples, a positive profit is achievable with only a single point deviation from the median in either direction (the confidence intervals indicated by error bars do not extend into negative values). Averaged across all $n = 21$ stratifications, the expected profit of a unit bet is $0.022 \pm 0.011$, $0.090 \pm 0.021$, and $0.15 \pm 0.030$ when the spread exceeds the median by 1, 2, and 3 points, respectively (mean ± standard deviation over $n = 21$ stratifications, each of which is an average over 1000 bootstrap ensembles). Similarly, the expected return is $0.023 \pm 0.013$, $0.089 \pm 0.026$, and $0.15 \pm 0.037$ when the spread undershoots the median by 1, 2, and 3 points respectively. This indicates that sportsbooks must estimate the median outcome with high precision in order to prevent the possibility of positive returns.

**Table 2. The relationship between the sportsbook's estimate of the point total and the actual total.** Matches were stratified into 24 subsamples defined by the value of the sportsbook total. The dependent variables are the 0.476, 0.5, and 0.524 quantiles of the true point total, as well as the expected profit of wagering conditioned on the amount of bias in the sportsbook's total.

| Total | | 0.476 | Median | 0.524 | $E\{\pi\|\tau = \bar{t} + k\}$ | | | | | | |
|---|---|---|---|---|---|---|---|---|---|---|---|
| | | | | | k = -3 | k = -2 | k = -1 | k = 0 | k = 1 | k = 2 | k = 3 |
| 37.0 | 0.025 | 33.966 | 34.866 | 35.846 | 0.057 | 0.020 | -0.013 | -0.045 | -0.013 | 0.019 | 0.051 |
| | 0.975 | 39.828 | 41.109 | 42.328 | 0.176 | 0.102 | 0.026 | -0.045 | 0.021 | 0.083 | 0.145 |
| | Mean | 36.602 | 37.761 | 38.992 | 0.115 | 0.060 | 0.006 | -0.045 | 0.003 | 0.049 | 0.094 |
| 37.5 | 0.025 | 35.624 | 36.647 | 37.626 | 0.080 | 0.037 | -0.005 | -0.045 | -0.004 | 0.037 | 0.076 |
| | 0.975 | 40.869 | 41.888 | 42.908 | 0.189 | 0.113 | 0.035 | -0.045 | 0.032 | 0.106 | 0.176 |
| | Mean | 38.169 | 39.153 | 40.154 | 0.132 | 0.074 | 0.014 | -0.045 | 0.013 | 0.070 | 0.125 |
| 38.0 | 0.025 | 34.386 | 35.505 | 36.586 | 0.079 | 0.037 | -0.004 | -0.045 | -0.003 | 0.042 | 0.089 |
| | 0.975 | 39.828 | 40.728 | 41.489 | 0.190 | 0.119 | 0.039 | -0.045 | 0.042 | 0.130 | 0.217 |
| | Mean | 37.281 | 38.237 | 39.149 | 0.131 | 0.075 | 0.016 | -0.045 | 0.020 | 0.086 | 0.153 |
| 39.0 | 0.025 | 32.706 | 33.566 | 34.606 | 0.033 | 0.006 | -0.020 | -0.045 | -0.021 | 0.006 | 0.034 |
| | 0.975 | 39.967 | 41.349 | 42.749 | 0.190 | 0.104 | 0.025 | -0.045 | 0.021 | 0.084 | 0.143 |
| | Mean | 36.070 | 37.420 | 38.794 | 0.098 | 0.047 | -0.000 | -0.045 | -0.001 | 0.042 | 0.086 |
| 39.5 | 0.025 | 34.926 | 35.904 | 36.766 | 0.082 | 0.039 | -0.004 | -0.045 | -0.005 | 0.038 | 0.083 |
| | 0.975 | 40.688 | 41.528 | 42.289 | 0.195 | 0.118 | 0.038 | -0.045 | 0.041 | 0.128 | 0.214 |
| | Mean | 37.820 | 38.772 | 39.712 | 0.138 | 0.078 | 0.017 | -0.045 | 0.018 | 0.082 | 0.146 |
| 40.0 | 0.025 | 37.967 | 38.807 | 39.607 | 0.108 | 0.057 | 0.006 | -0.045 | 0.005 | 0.053 | 0.098 |
| | 0.975 | 41.629 | 42.529 | 43.588 | 0.205 | 0.125 | 0.042 | -0.045 | 0.043 | 0.128 | 0.204 |
| | Mean | 39.882 | 40.719 | 41.559 | 0.156 | 0.092 | 0.024 | -0.045 | 0.024 | 0.090 | 0.151 |
| 40.5 | 0.025 | 38.967 | 39.827 | 40.707 | 0.099 | 0.049 | 0.001 | -0.045 | -0.002 | 0.041 | 0.082 |
| | 0.975 | 43.049 | 44.149 | 45.289 | 0.205 | 0.124 | 0.040 | -0.045 | 0.037 | 0.113 | 0.184 |
| | Mean | 40.948 | 41.835 | 42.764 | 0.153 | 0.088 | 0.021 | -0.045 | 0.018 | 0.076 | 0.130 |
| 41.0 | 0.025 | 38.327 | 39.287 | 40.288 | 0.094 | 0.047 | 0.000 | -0.045 | 0.001 | 0.048 | 0.098 |
| | 0.975 | 42.048 | 42.928 | 43.829 | 0.172 | 0.101 | 0.029 | -0.045 | 0.030 | 0.106 | 0.182 |
| | Mean | 40.232 | 41.203 | 42.158 | 0.132 | 0.073 | 0.014 | -0.045 | 0.015 | 0.077 | 0.139 |
| 41.5 | 0.025 | 40.128 | 41.048 | 41.987 | 0.095 | 0.047 | 0.000 | -0.045 | -0.003 | 0.036 | 0.074 |
| | 0.975 | 44.269 | 45.269 | 46.410 | 0.192 | 0.114 | 0.035 | -0.045 | 0.035 | 0.110 | 0.176 |
| | Mean | 42.202 | 43.130 | 44.095 | 0.144 | 0.081 | 0.018 | -0.045 | 0.015 | 0.072 | 0.124 |
| 42.0 | 0.025 | 39.306 | 40.567 | 41.927 | 0.061 | 0.025 | -0.010 | -0.045 | -0.010 | 0.025 | 0.059 |
| | 0.975 | 44.529 | 45.769 | 47.031 | 0.133 | 0.077 | 0.017 | -0.045 | 0.018 | 0.081 | 0.140 |
| | Mean | 41.914 | 43.131 | 44.330 | 0.095 | 0.050 | 0.003 | -0.045 | 0.003 | 0.052 | 0.100 |
| 42.5 | 0.025 | 38.507 | 39.547 | 40.428 | 0.102 | 0.053 | 0.004 | -0.045 | 0.004 | 0.054 | 0.099 |
| | 0.975 | 42.488 | 43.349 | 44.289 | 0.201 | 0.126 | 0.043 | -0.045 | 0.042 | 0.126 | 0.203 |
| | Mean | 40.554 | 41.408 | 42.257 | 0.150 | 0.088 | 0.023 | -0.045 | 0.023 | 0.090 | 0.153 |
| 43.0 | 0.025 | 41.327 | 42.308 | 43.228 | 0.096 | 0.051 | 0.004 | -0.045 | 0.005 | 0.056 | 0.106 |
| | 0.975 | 44.629 | 45.549 | 46.469 | 0.173 | 0.104 | 0.031 | -0.045 | 0.033 | 0.110 | 0.183 |
| | Mean | 43.073 | 44.004 | 44.920 | 0.133 | 0.077 | 0.017 | -0.045 | 0.018 | 0.081 | 0.142 |
| 43.5 | 0.025 | 40.407 | 41.167 | 41.907 | 0.120 | 0.061 | 0.005 | -0.045 | 0.002 | 0.046 | 0.088 |
| | 0.975 | 43.548 | 44.549 | 45.629 | 0.230 | 0.139 | 0.045 | -0.045 | 0.036 | 0.110 | 0.177 |
| | Mean | 41.898 | 42.740 | 43.654 | 0.174 | 0.099 | 0.025 | -0.045 | 0.018 | 0.076 | 0.130 |
| 44.0 | 0.025 | 41.347 | 42.328 | 43.288 | 0.097 | 0.051 | 0.003 | -0.045 | 0.004 | 0.053 | 0.103 |
| | 0.975 | 44.789 | 45.709 | 46.669 | 0.164 | 0.099 | 0.028 | -0.045 | 0.028 | 0.100 | 0.170 |
| | Mean | 43.185 | 44.140 | 45.090 | 0.131 | 0.074 | 0.015 | -0.045 | 0.015 | 0.075 | 0.134 |
| 44.5 | 0.025 | 40.508 | 41.487 | 42.527 | 0.084 | 0.039 | -0.004 | -0.045 | -0.005 | 0.034 | 0.076 |
| | 0.975 | 44.589 | 45.749 | 46.890 | 0.182 | 0.105 | 0.028 | -0.045 | 0.024 | 0.087 | 0.150 |
| | Mean | 42.545 | 43.575 | 44.660 | 0.132 | 0.071 | 0.011 | -0.045 | 0.008 | 0.060 | 0.112 |

*(Continued)*

**Table 2.** (Continued)

| Total | | 0.476 | Median | 0.524 | $E\{\pi\|\tau=\bar{t}+k\}$ | | | | | | |
|---|---|---|---|---|---|---|---|---|---|---|---|
| | | | | | k = -3 | k = -2 | k = -1 | k = 0 | k = 1 | k = 2 | k = 3 |
| 45.0 | 0.025 | 42.727 | 43.667 | 44.569 | 0.100 | 0.053 | 0.004 | -0.045 | 0.003 | 0.051 | 0.097 |
| | 0.975 | 46.409 | 47.290 | 48.351 | 0.181 | 0.109 | 0.033 | -0.045 | 0.033 | 0.106 | 0.175 |
| | Mean | 44.553 | 45.461 | 46.379 | 0.141 | 0.081 | 0.019 | -0.045 | 0.018 | 0.079 | 0.137 |
| 45.5 | 0.025 | 41.268 | 42.268 | 43.188 | 0.086 | 0.043 | -0.000 | -0.045 | 0.001 | 0.051 | 0.099 |
| | 0.975 | 45.729 | 46.549 | 47.430 | 0.189 | 0.118 | 0.039 | -0.045 | 0.042 | 0.128 | 0.204 |
| | Mean | 43.593 | 44.514 | 45.403 | 0.136 | 0.079 | 0.018 | -0.045 | 0.021 | 0.088 | 0.153 |
| 46.0 | 0.025 | 41.107 | 42.248 | 43.308 | 0.072 | 0.031 | -0.008 | -0.045 | -0.010 | 0.023 | 0.056 |
| | 0.975 | 45.529 | 46.809 | 48.131 | 0.154 | 0.091 | 0.024 | -0.045 | 0.022 | 0.084 | 0.142 |
| | Mean | 43.342 | 44.452 | 45.620 | 0.112 | 0.060 | 0.007 | -0.045 | 0.005 | 0.053 | 0.097 |
| 46.5 | 0.025 | 43.068 | 43.788 | 44.609 | 0.115 | 0.060 | 0.007 | -0.045 | 0.004 | 0.052 | 0.096 |
| | 0.975 | 47.029 | 47.949 | 48.950 | 0.222 | 0.135 | 0.044 | -0.045 | 0.041 | 0.122 | 0.198 |
| | Mean | 44.986 | 45.813 | 46.671 | 0.168 | 0.098 | 0.025 | -0.045 | 0.022 | 0.086 | 0.146 |
| 47.0 | 0.025 | 40.887 | 41.707 | 42.548 | 0.108 | 0.057 | 0.006 | -0.045 | 0.005 | 0.052 | 0.097 |
| | 0.975 | 44.329 | 45.289 | 46.269 | 0.196 | 0.119 | 0.037 | -0.045 | 0.036 | 0.112 | 0.181 |
| | Mean | 42.593 | 43.467 | 44.364 | 0.152 | 0.088 | 0.021 | -0.045 | 0.019 | 0.080 | 0.137 |
| 47.5 | 0.025 | 44.828 | 45.848 | 46.789 | 0.070 | 0.030 | -0.008 | -0.045 | -0.009 | 0.030 | 0.073 |
| | 0.975 | 50.411 | 51.271 | 52.091 | 0.171 | 0.099 | 0.028 | -0.045 | 0.033 | 0.117 | 0.203 |
| | Mean | 47.449 | 48.523 | 49.579 | 0.120 | 0.064 | 0.009 | -0.045 | 0.011 | 0.069 | 0.132 |
| 48.0 | 0.025 | 44.909 | 45.688 | 46.409 | 0.112 | 0.057 | 0.004 | -0.045 | 0.002 | 0.046 | 0.088 |
| | 0.975 | 48.851 | 49.890 | 50.931 | 0.215 | 0.132 | 0.042 | -0.045 | 0.038 | 0.116 | 0.185 |
| | Mean | 46.792 | 47.651 | 48.552 | 0.163 | 0.093 | 0.023 | -0.045 | 0.019 | 0.080 | 0.136 |
| 48.5 | 0.025 | 44.389 | 45.248 | 46.248 | 0.079 | 0.036 | -0.005 | -0.045 | -0.006 | 0.037 | 0.083 |
| | 0.975 | 49.852 | 50.731 | 51.571 | 0.199 | 0.113 | 0.033 | -0.045 | 0.035 | 0.119 | 0.206 |
| | Mean | 47.036 | 48.066 | 49.071 | 0.130 | 0.069 | 0.012 | -0.045 | 0.013 | 0.075 | 0.138 |
| 49.0 | 0.025 | 44.988 | 45.968 | 47.049 | 0.048 | 0.015 | -0.016 | -0.045 | -0.016 | 0.014 | 0.043 |
| | 0.975 | 51.651 | 52.971 | 54.112 | 0.182 | 0.101 | 0.027 | -0.045 | 0.023 | 0.089 | 0.152 |
| | Mean | 48.087 | 49.320 | 50.596 | 0.107 | 0.054 | 0.003 | -0.045 | 0.002 | 0.048 | 0.094 |

The analysis was repeated on the data of point totals. A deviation from the true median of only 1 point was sufficient to permit a positive expected profit in all four of the largest stratifications (Fig 5; $t \in \{41, 43, 44, 45\}$; error bars indicate 95% confidence intervals; data for all samples is provided in Table 2). When the sportsbook overestimates the median total by 1, 2, and 3 points, the expected profit on a unit bet is $0.014 \pm 0.0071$, $0.073 \pm 0.014$, and $0.13 \pm 0.020$, respectively (mean ± standard deviation over $n = 24$ samples, each of which is a average over 1000 bootstrap resamples). When the sportsbook underestimates the median, the expected profit on a unit bet is $0.015 \pm 0.0071$, $0.076 \pm 0.014$, and $0.14 \pm 0.020$, for deviations of 1, 2, and 3 points, respectively. Note that despite the dependent variable having a larger magnitude (compared to margin of victory), the required sportsbook error to permit positive profit is the same as shown by the analysis of point spreads.

## Discussion

The theoretical results presented here, despite seemingly straightforward, have eluded explication in the literature. The central message is that optimal wagering on sports requires accurate estimation of the outcome variable's quantiles. For the two most common types of bets—point spread and point total—estimation of the 0.476, 0.5 (median), and 0.524 quantiles constitutes

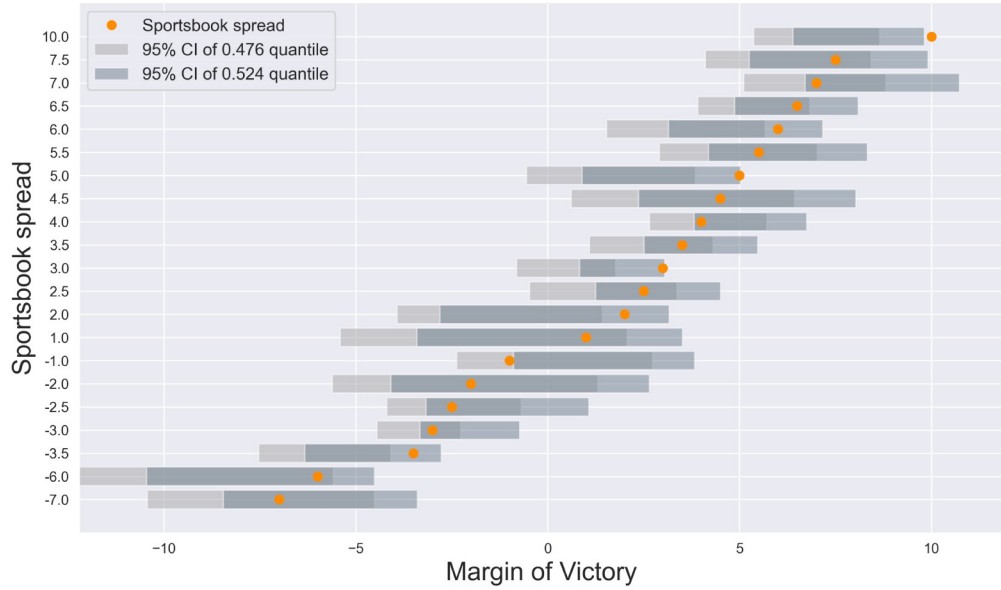

**Fig 2. Do sportsbook point spreads deviate from the 0.476-0.524 quantiles?** With a standard payout of $\phi = 0.91$, achieving a positive expected profit is only feasible if the sportsbook point spread falls outside of the 0.476-0.524 quantiles of the margin of victory. The 0.476 and 0.524 quantiles were thus estimated for each stratified sample of NFL matches. Light (dark) black bars indicate the 95% confidence intervals of the 0.476 (0.524) quantiles. Orange markers indicate the sportsbook point spread, which fell within the quantile confidence intervals for the large majority of stratifications. An exception was $s = 5$, where the sportsbook appeared to overestimate the margin of victory. For two other stratifications ($s = 3$ and $s = 10$), the 0.524 quantile tended to underestimate the sportsbook spread, with the 95% confidence intervals extending to just above the spread.

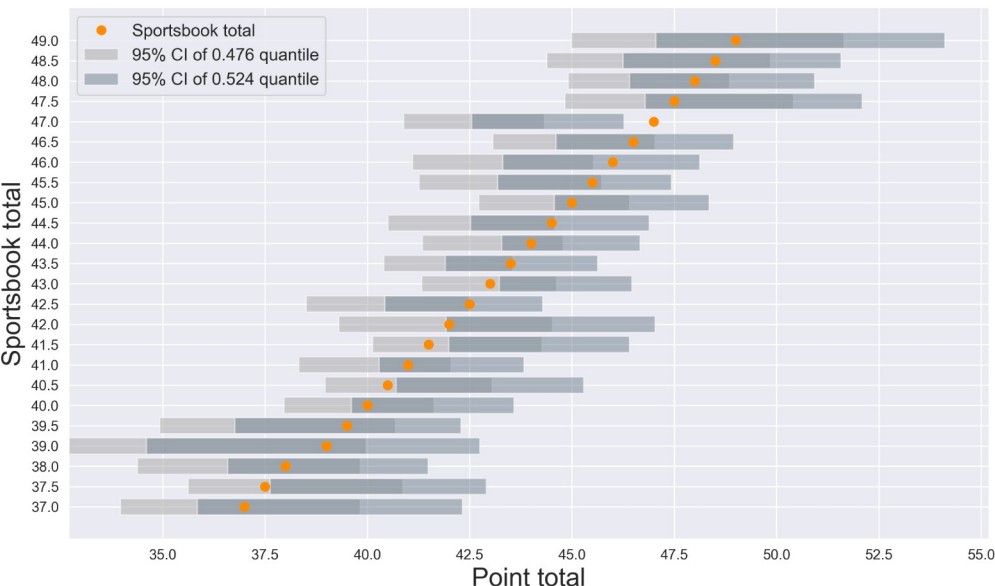

**Fig 3. Do sportsbook point totals deviate from the 0.476-0.524 interval?** The 0.476 and 0.524 quantiles of the true *point total* were estimated for each stratified sample of NFL matches. For all but one stratification ($t = 47$, 95% confidence interval [41.59-45.42], sportsbook overestimates the total), the confidence intervals of the sample quantiles contained the sportsbook proposition. Visual inspection of the data suggests that, in the NFL betting market at least, sportsbooks are very adept at proposing totals that fall within the critical 0.476-0.524 quantiles.

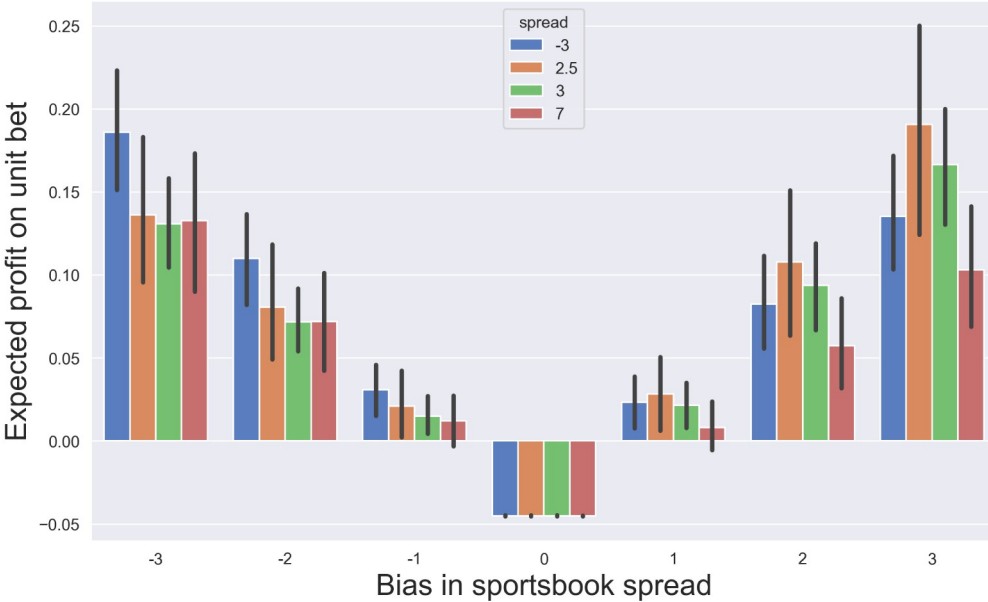

**Fig 4. How large of a bias in the point spread is required for positive expected profit?** In order to estimate the magnitude of the deviation between sportsbook point spread and median margin of victory that is required to permit a positive profit to the bettor, the *hypothetical* expected profit was computed for point spreads that differ from the true median by 1, 2, and 3 points in each direction. The analysis was performed separately within each stratified sample, and the figure shows the results of the four largest samples. For 3 of the 4 stratifications, a sportsbook bias of only a single point is required to permit a positive expected return (height of the bar indicates the expected profit of a unit bet assuming that the bettor wagers on the side with the higher probability of winning; error bars indicate the 95% confidence intervals as computed with the bootstrap). For a sportsbook spread of $s = 3$ (dark black bars), the expected profit on a unit bet is 0.021 [0.008-0.035], 0.094 [0.067-0.119], and 0.166 [0.13-0.2] when the sportsbook's bias is +1, +2, and +3 points, respectively (mean and confidence interval over 500 bootstrap resamples).

the primary task of the bettor (assuming a standard commission of 4.5%). For a given match, the bettor must compare the estimated quantiles to the sportsbook's proposed value, and first decide whether or not to wager (*Theorem 2*), and if so, on which side (*Theorem 1*).

The sportsbook's proposed spread (or point total) effectively delineates the potential outcomes for the bettor (*Theorem 3*). For a standard commission of 4.5%, the result is that if the sportsbook produces an estimate within 2.4 percentiles of the true median outcome, wagering always yields a negative expected profit—*even if consistently wagering on the side with the higher probability of winning the bet*. This finding underscores the importance of not wagering on matches in which the sportsbook has accurately captured the median outcome with their proposition. In such matches, the minimum error rate is lower bounded by 47.6%, the maximum error rate is upper bounded by 52.4%, and the excess error rate (*Theorem 4*) is upper bounded by 4.8%.

The seminal findings of Kuypers [13] and Levitt [23], however, imply that sportsbooks may sometimes deliberately propose values that deviate from their estimated median to entice a preponderance of bets on the side that maximizes excess error. For example, by proposing a point spread that exaggerates the median margin of victory of a home favorite, the minimum error rate may become, for example, 45% (when wagering on the road team), and the excess error rate when wagering on the home team is 10%. In this hypothetical scenario, the sportsbook may predict that, due to the public's bias for home favorites, a majority of the bets will be placed on the home team. The empirical data presented here hint at this phenomenon, and are in alignment with previous reports of market inefficiencies in the NFL betting market [5, 32–

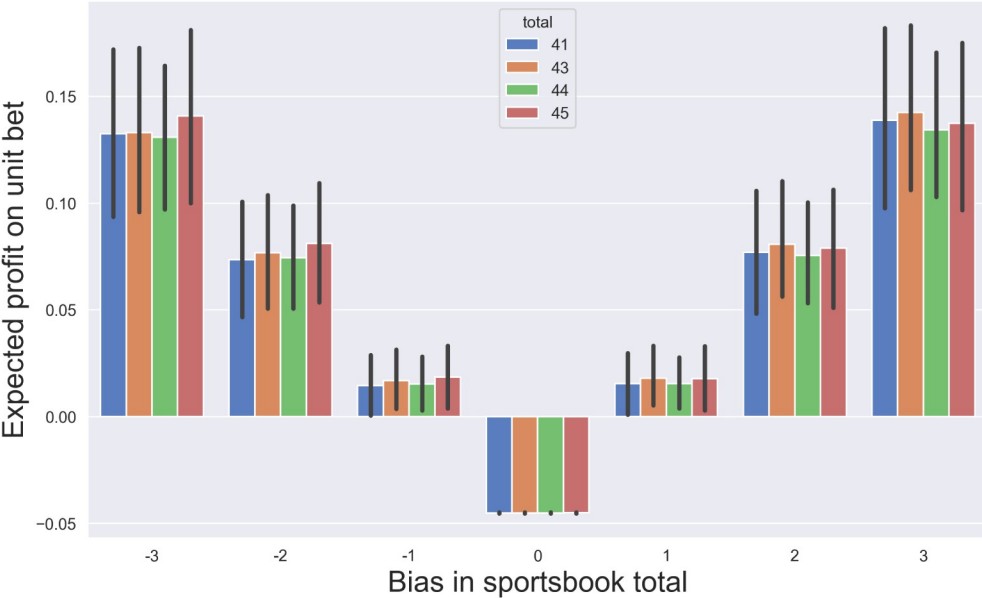

**Fig 5. How large of a bias in the point total is required for positive expected profit?** Vertical axis depicts the expected profit of an over-under wager, conditioned on the sportsbook's posted total deviating from the true margin by a value of 1, 2, or 3 points (horizontal axis). The analysis was performed separately for each unique sportsbook total, and the figure displays the results for the four largest samples. A deviation from the true median of a single point permits a positive expected profit in all four of the depicted groups. For a sportsbook total of $t = 44$ (green bars), the expected profit on a unit bet is 0.015 [0.004-0.028], 0.075 [0.053-0.10], and 0.13 [0.10-0.17] when the sportsbook's bias is +1, +2, and +3 points, respectively (mean and confidence interval over 500 bootstrap resamples).

35]. Namely, the sportsbook point spread was found to slightly overestimate the median margin of victory for some subsets of the data (Fig 2). Indeed, the stratifications showing this trend were home favorites, agreeing with the idea that the sportsbooks are exploiting the public's bias for wagering on the favorite [23].

The analysis of sportsbook point spreads performed here indicates that only a single point deviation from the true median is sufficient to allow one of the betting options to yield a positive expectation. On the other hand, realization of this potential profit requires that the bettor correctly, and systematically, identify the side with the higher probability of winning against the spread. Forecasting the outcomes of sports matches against the spread has been elusive for both experts and models [6, 36]. Due to the abundance of historical data and user-friendly statistical software packages, the employment of quantitative modeling to aid decision-making in sports wagering [37] is strongly encouraged. The following suggestions are aimed at guiding model-driven efforts to forecast sports outcomes.

## The argument against binary classification for sports wagering

The minimum error and minimum excess error rates defined in Theorems 3 and 4, respectively, are analogous to the Bayes' minimum risk and Bayes' excess risk in binary classification [38]. Indeed, one can cast the estimation of margin of victory in sports wagering as a binary classification problem, aiming to predict the event of "the home team winning against the spread". Here this approach is not advocated. In conventional binary classification, the target variable (or "class label") is static and assumed to represent some phenomenon (e.g. presence or absence of an object). In the context of sports wagering, however, the event $m > s$ need not

be uniform for different matches. For example, the event of a large home favorite winning against the spread may differ qualitatively from that of a small home "underdog" winning against the spread. Moreover, the sportsbook's proposed point spread is a dynamic quantity. To illustrate the potential difficulty of utilizing classifiers in sports wagering, consider the case of a match with a posted spread of $s = 4$, where the goal is to predict the sign of $m - 4$. But now imagine that the the spread moves to $s = 3$. The resulting binary classification problem is now to predict the sign of $m - 3$, and it is not straightforward to adapt the previously constructed classifier to this new problem setting. One may be tempted to modify the bias term of the classifier, but it is unclear by how much it should be adjusted, and also whether a threshold adjustment is in fact the optimal approach in this scenario. On the other hand, by posing the problem as a regression, it is trivial to adapt one's optimal decision: the output of the regression can simply be compared to the new spread.

## The case for quantile regression

Conventional ordinary least-squares (OLS) regression yields estimates of the mean of a random variable, conditioned on the predictors. This is achieved by minimizing the mean squared error between the predicted and target variable.

The findings presented here suggest that conventional regression may be a sub-optimal approach to guiding wagering decisions, whose optimality relies on knowledge of the median and other quantiles. The presence of outliers and multi-modal distributions, as may be expected in sports outcomes, increases the deviation between the mean and median of a random variable. In this case, the dependent variable of conventional regression is distinct from the median and thus less relevant to the decision-making of the sports bettor. The significance of this may be exacerbated by the high noise level on the target random variable, and the low ceiling on model accuracy that this imposes.

Therefore, a more suitable approach to quantitative modeling in sports wagering is to employ quantile regression, which estimates a random variable's quantiles by minimizing the quantile loss function [39]. Any features that are expected to forecast sports outcomes could be provided as the predictors in a quantile regression to produce estimates that are aligned with the bettor's objectives: to avoid wagering on matches with negative expectation for both outcomes, and to wager on the side with zero excess error.

## Potential challenges in moneyline wagering

Optimal wagering requires knowledge of the $\left(\frac{\phi_h}{1+\phi_h}\right)$ and $\left(\frac{1}{1+\phi_v}\right)$ quantiles of the outcome variable. For point spread and point total wagers, the values of $\phi_h$ and $\phi_v$ do not substantially vary across matches. As a result, one can train a model on historical data to generate estimates of these canonical quantiles for future matches. Alternatively, one can develop a model to estimate the median and utilize it in conjunction with knowledge of how many points represents the requisite 2.4 percentile deviation. However, in the case of moneyline wagering, the payouts $\phi_h$ and $\phi_v$ do vary greatly across matches, meaning that one needs to estimate variable quantiles for different matches. This poses a challenge to predictive modeling for moneyline wagering, which will require estimating either very many quantiles or the entire distribution of the outcome variable. This seems to suggest a potential advantage of point spread and point total wagering: quantitative models can be trained to predict one or a few nominal quantiles, without the need to estimate the entire distribution of the outcome variable.

## Bias-variance in sports wagering

One may intuit that the goal of the sports bettor is to produce a closer estimate of the median outcome than the sportsbook. However, an important consequence of *Theorem 4* is that estimators of the median outcome in sports betting need not be more precise than the sportsbook's proposition in order to achieve a positive expected profit. Rather, the goal of the statistical model is to produce estimates that yield sampling distributions with mass on the same side of the sportsbook proposition as the true median. Variations on this fundamental result have been previously presented in [28, 40], which show that suboptimal models—those that yield estimates that deviate substantially from the true outcome—are in fact capable of *systematically* generating positive returns. In statistical terms, the optimal estimator should be permitted to exhibit a large bias such that its degrees of freedom can be utilized to identify the sign of $\bar{m} - s$, regardless of how close the estimate $\hat{m}$ is to the true median. In the event that the estimate falls on the "correct" side of the spread, a low estimator variance will minimize the excess error rate. Interestingly, for a fixed estimator variance, the excess error in this case is minimized with an infinite bias.

The view that low variance implies "simple" models has recently been challenged in the context of artificial neural networks [41]. Nevertheless, the desire for low-variance, high-bias modeling in sports wagering does suggest the preference for simpler models. Thus, it is advocated to employ a limited set of predictors and a limited capacity of the model architecture. This is expected to translate to improved generalization to future data.

## Sport-specific considerations

The three types of wagers considered in this work—point spread, moneyline, and over-under—are the most popular bet types in North American sports. The empirical analysis employed data from the National Football League (NFL). One unique aspect of American football is its scoring system, in which the points accumulated by each team increase primarily in increments of 3 or 7 points. The structure of the scoring imposes constraints on the distribution of the margin of victory $m$. For example, in American Football, the distribution of the margin of victory is expected to exhibit local maxima near values such as: ±3, ±7, ±10. In the case of games in the National Basketball Association (NBA), the most common margins of victory tend to occur in the 5-10 interval, reflecting the overall higher point totals in basketball and its most common point increments (2 and 3). As a result, the shape and quantiles of the distribution of $m$ may vary qualitatively between the NBA and NFL.

As a final illustrative example of the importance of the quantiles of $m$, consider the hypothetical scenario of two American football teams playing a match whose parameters $\theta$ have been exactly matched three times previously. In those past matches, the outcomes were $m = 3$, $m = 7$, and $m = 35$. In this fictitious example, the median is 7 but the mean is 15. Now imagine that the point spread for the next match has been set to $s = 10$ (home team favored to win the match by 10 points). Assuming that one has committed to wagering on the match, the optimal decision is to bet on the visiting team, despite that fact that the home team has won the previous matches by an average of 15 points.

## Materials and methods

All analysis was performed with custom Python code compiled into a Jupyter Notebook (available at https://github.com/dmochow/optimal_betting_theory). The figures and tables in this manuscript may be reproduced by executing the notebook.

### Empirical data

Historical data from the National Football League (NFL) was obtained from bettingdata.com, who has courteously permitted the data to be shared on the repository listed above. All regular season matches from 2002 to 2022 were included in the analysis ($n$ = 5412). The data set includes point spreads and point totals (with associated payouts) from a variety of sportsbooks, as well as a "consensus" value. The latter was utilized for all analysis.

### Data stratification

In order to estimate quantiles of the distributions of margin of victory and point totals from heterogeneous data (i.e., matches with disparate relative strengths of the home and visiting teams), the sportsbook point spread and sportsbook point total were used as a surrogate for the parameter vector defining the identity of each individual match ($\theta$ in the text). This permitted the estimation of the 0.476, 0.5, and 0.524 quantiles over subsets of congruent matches.

Only spreads or totals with at least 100 matches in the dataset were included, such that estimation of the median would be sufficiently reliable. To that end, data was stratified into 21 samples for the analysis of margin of victory: {-7, -6, -3.5, -3, -2.5, -2, -1, 1, 2, 2.5, 3, 3.5, 4, 4.5, 5, 5.5, 6, 6.5, 7, 7.5, 10) and 24 samples for the analysis of point totals (37, 37.5, 38, 39, 39.5, 40, 40.5, 41, 41.5, 42, 42.5, 43, 43.5, 44, 44.5, 45, 45.5, 46, 46.5, 47, 47.5, 48, 48.5, 49 }. This resulted in the employment of $n$ = 3843 matches in the analysis of point spreads and $n$ = 4300 matches in the analysis of point totals.

Note that the stratification process did not account for varying payouts, for example −110 versus −105 in the American odds system, as this would greatly increase the number of stratified samples while decreasing the number of matches in each sample. It is likely that the resulting error is negligible, however, due to the likelihood of the payout discrepancy being fairly balanced across the home and visiting teams.

### Median estimation

In order to overcome the discrete nature of the margins of victory and point totals, kernel density estimation was employed to produce continuous quantile estimates. The *KernelDensity* function from the *scikit-learn* software library was employed with a Gaussian kernel and a bandwidth parameter of 2. For the margin of victory, the density was estimated over 4000 points ranging from -40 to 40. For the analysis of point totals, the density was estimated over 4000 points ranging from 10 to 90. The regression analysis relating median outcome to sportsbook estimates (Fig 1) was performed with ordinary least squares (OLS).

### Confidence interval estimation

In order to generate variability estimates for the 0.476, 0.5, and 0.524 quantiles of the margin of victory and point total, the bootstrap [42] technique was employed. 1000 resamples of the same size as the original sample were generated in each case. The confidence intervals were then constructed as the interval between the 2.5 and 97.5 percentiles of the relevant quantity. Bootstrap resampling was also employed to derive confidence intervals on the regression parameters relating the median outcomes to sportsbook spreads or totals (Fig 1), as well as the confidence intervals on the expected profit of wagering conditioned on a fixed sportsbook bias (Figs 4 and 5).

### Expected profit estimation

To quantify the relationship between a sportsbook bias and the associated upper bound on wagering performance, the empirical CDF of each stratified sample was converted into an

expected profit, conditioned on a hypothetical spread (or total) that deviated from the true median by fixed increments of -3, -2, -1, 0, 1, 2, and 3 points. More specifically, the expected values were first computed separately for the case of wagering on the home and visiting teams:

$$
\begin{aligned}
E\{\pi|\text{bet home}\} &= \phi_h - \hat{F}_m(s^*)(1 + \phi_h), \\
E\{\pi|\text{bet visitor}\} &= \hat{F}_m(s^*)(\phi_v + 1) - 1,
\end{aligned}
\tag{15}
$$

where $\phi_h$ and $\phi_v$ were set to $100/110 = 0.91$, and where $\hat{F}_m(s^*)$ denotes the kernel density estimate of the CDF of margin of victory evaluated at the hypothesized spread $s^*$:

$$
s^* = \bar{m} + k, \quad k \in \{-3, -2, -1, 0, 1, 2, 3\},
$$

where $\bar{m}$ is the median margin of victory as computed on the stratified sample of matches and $k$ is the hypothesized sportsbook bias.

To model the idealized case of always placing the wager on the side with the higher probability of winning against the spread, the reported expected profit was taken as the maximum of the two expected values in (15). The analogous procedure was conducted for the analysis of point totals.

## Acknowledgments

The author would like to thank Ed Miller and Mark Broadie for fruitful discussions during the preparation of the manuscript. The author would also like to acknowledge the effort of the reviewers, in particular Fabian Wunderlich, for providing many helpful comments and critiques throughout peer review.

## Author Contributions

**Conceptualization:** Jacek P. Dmochowski.

**Data curation:** Jacek P. Dmochowski.

**Formal analysis:** Jacek P. Dmochowski.

**Investigation:** Jacek P. Dmochowski.

**Methodology:** Jacek P. Dmochowski.

**Visualization:** Jacek P. Dmochowski.

**Writing – original draft:** Jacek P. Dmochowski.

**Writing – review & editing:** Jacek P. Dmochowski.

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
