## [Decision Letter · Decision Letter 0]

1 Feb 2023

PONE-D-22-34727

Statistical principles of optimal decision-making in sports wagering

PLOS ONE

Dear Dr. Dmochowski,

Thank you for submitting your manuscript to PLOS ONE. After careful consideration, we have decided that your manuscript does not meet our criteria for publication and must therefore be rejected.

Specifically:

I am sorry that we cannot be more positive on this occasion, but hope that you appreciate the reasons for this decision.

Kind regards,

Olivier Bos

Academic Editor

PLOS ONE

Additional Editor Comments:

Dear author,

I have now heard back from an expert reviewer on your paper titled "Statistical Principles of Optimal Decision-Making in Sports Wagering." The reviewer expressed that the work does not adequately add to the existing literature on this topic. Given her/his expertise in the field I have decided to follow her/his recommendation.

The reviewer provided comments and suggestions which I believe will be valuable in improving your work and potentially finding a more suitable outlet for publication.

I regret to inform you of this disappointing decision. I wish you the best of luck with your research and future publications.

Sincerely,

Olivier Bos

Reviewers' comments:

Reviewer's Responses to Questions

**Comments to the Author**

1. Is the manuscript technically sound, and do the data support the conclusions?

Reviewer #1: Partly

2. Has the statistical analysis been performed appropriately and rigorously? 

Reviewer #1: No

3. Have the authors made all data underlying the findings in their manuscript fully available?

Reviewer #1: Yes

4. Is the manuscript presented in an intelligible fashion and written in standard English?

Reviewer #1: Yes

5. Review Comments to the Author

Reviewer #1: Report on “Statistical Principles of Optimal Decision-Making in Sports Wagering”

This paper analyzes the efficiency of betting markets for American professional football games from 2002-2022. The analysis includes bets on the margin of victory (point spread) and on the total number of points scored.

In each case, bets are analyzed that are approximately even money bets. A casino specifies a point spread or point total, and the bettor can either choose the team to bet on (in the case of a point spread) or choose to bet that the total will higher or lower than the specified “over/under”. The most common arrangement is that a bettor wins the amount bet if they win but loses 1.1 times that amount if they lose. Therefore a bettor would need to win 11/21 = 0.524 or more of their bets over the long run to earn positive profits.

The analysis is conducted non-parametrically. Games are placed in bins according to their point spread and again according to their point totals. Assuming that a bin ends up containing 100 games or more, the paper calculates the 47.6th and 52.4th percentiles of the outcome distribution and compares them to the “consensus” point spread or point total offered by the casinos. This is equivalent to calculating the mean binary outcome for bets in each direction for the bin and comparing it to the profitability thresholds of 0.524 or 1 – 0.524 = 0.476.

The paper finds that many bins have sample means that are outside the 0.476 – 0.524 range (see Figure 2). It highlights this finding in the abstract: “approximately two-thirds of matches permit a positive expected profit (but only 43% for bets on point total).” Of course, sample means will vary around population means; with 100 binary outcomes, we would expect the difference between the sample mean and population mean to be approximately 0.5(1 – 0.5)/sqrt(100) = 0.05. There is no analysis of whether we can reject a null hypothesis that the population means fall within the 0.476 – 0.524 interval.

Put another way, in order for a bettor to have earned a positive profit, they would have had to know in advance of the sample period which specific bins would yield outcome probabilities below 0.476 or above 0.524. The statement that “approximately two-thirds of matches permit a positive expected profit” assumes an ability to cherry pick bins after the fact.

The general patterns in Figure 1, that home underdogs perform better relative to the point spread than home favorites and that outcomes tend to be less extreme than predicted by extreme over/unders, are well-documented in the literature. The literature in this area is quite vast (search “NFL betting” in Google Scholar to see what I mean). I do not find the incremental contribution of this paper to be compelling enough for PLOS-ONE.

In this context, I should also mention that I find the claim in the abstract that “here it is shown that sports wagering is effectively a problem of quantile estimation, with the median outcome having a crucial role in optimal decision making” to be a bit strange. That the probability of an outcome being above or below a specified point spread or total, and therefore the quantiles of the distribution of the point spread or total, is central to the profitability of sports betting strikes me as completely obvious to everyone.

6. PLOS authors have the option to publish the peer review history of their article (what does this mean?). If published, this will include your full peer review and any attached files.

Reviewer #1: No

- - - - -

---

## [Author Response · Author response to Decision Letter 0]

7 Mar 2023

Please see the "Response to Reviewers" document.

---

## [Decision Letter · Decision Letter 1]

24 Apr 2023

PONE-D-22-34727R1A statistical theory of optimal decision-making in sports wageringPLOS ONE

Dear Dr. Dmochowski,

Thank you for submitting your manuscript to PLOS ONE. After careful consideration, we feel that it has merit but does not fully meet PLOS ONE’s publication criteria as it currently stands. Therefore, we invite you to submit a revised version of the manuscript that addresses the points raised during the review process.

We recommend that it should be revised by taking into account the changes requested by Reviewers. I want to give you a chance to revise your manuscript. The Academic Editor will only review the manuscript in the next round to speed the review process.

We look forward to receiving your revised manuscript.

Kind regards,

Baogui Xin, Ph.D.

Academic Editor

PLOS ONE

Journal Requirements:

1. Please ensure that your manuscript meets PLOS ONE's style requirements, including those for file naming. The PLOS ONE style templates can be found athttps://journals.plos.org/plosone/s/file?id=wjVg/PLOSOne_formatting_sample_main_body.pdf and https://journals.plos.org/plosone/s/file?id=ba62/PLOSOne_formatting_sample_title_authors_affiliations.pdf.

4. We note that you have indicated that data from this study are available upon request. PLOS only allows data to be available upon request if there are legal or ethical restrictions on sharing data publicly. For more information on unacceptable data access restrictions, please see http://journals.plos.org/plosone/s/data-availability#loc-unacceptable-data-access-restrictions.In your revised cover letter, please address the following prompts:a) If there are ethical or legal restrictions on sharing a de-identified data set, please explain them in detail (e.g., data contain potentially sensitive information, data are owned by a third-party organization, etc.) and who has imposed them (e.g., an ethics committee). Please also provide contact information for a data access committee, ethics committee, or other institutional body to which data requests may be sent.b) If there are no restrictions, please upload the minimal anonymized data set necessary to replicate your study findings as either Supporting Information files or to a stable, public repository and provide us with the relevant URLs, DOIs, or accession numbers. For a list of acceptable repositories, please see http://journals.plos.org/plosone/s/data-availability#loc-recommended-repositories.We will update your Data Availability statement on your behalf to reflect the information you provide.

5. Please update your submission to use the PLOS LaTeX template. The template and more information on our requirements for LaTeX submissions can be found at http://journals.plos.org/plosone/s/latex.

Additional Editor Comments (if provided):

Reviewers' comments:

Reviewer's Responses to Questions

**Comments to the Author**

1. If the authors have adequately addressed your comments raised in a previous round of review and you feel that this manuscript is now acceptable for publication, you may indicate that here to bypass the “Comments to the Author” section, enter your conflict of interest statement in the “Confidential to Editor” section, and submit your "Accept" recommendation.

Reviewer #2: All comments have been addressed

Reviewer #3: (No Response)

2. Is the manuscript technically sound, and do the data support the conclusions?

Reviewer #2: Yes

Reviewer #3: Yes

3. Has the statistical analysis been performed appropriately and rigorously? 

Reviewer #2: Yes

Reviewer #3: Yes

4. Have the authors made all data underlying the findings in their manuscript fully available?

Reviewer #2: Yes

Reviewer #3: Yes

5. Is the manuscript presented in an intelligible fashion and written in standard English?

Reviewer #2: Yes

Reviewer #3: Yes

6. Review Comments to the Author

Reviewer #2: Please remove the heading "Results" prior to problem formulation. All the subheadings with a question mark should be revised as normal heading (page 6-7). Subsections should be named without any a punctuation mark (all over the paper). Materials and methods should be placed as appendix. Discussion should be named as discussion and conclusion.

Reviewer #3: For a better overview, please find my comments also attached as pdf.

Review PONE-D-22-34727R1

Thank you for giving me the opportunity to review this manuscript. I would like to underline that – in my opinion – the manuscript has merit and I am very confident that it will be worth publishing if several revisions are made to the manuscript.

Positive aspects

I will only very briefly mention the positive aspects of the manuscript as this review is intended to put more effort on possibilities for improvement. However, I would like to underline that I really enjoyed reading the manuscript. I like the fact that theoretical considerations are combined with empirical data. Moreover, it is interesting to see a manuscript in the domain of forecasting/sports betting that is able to answer relevant questions without the use of a concrete forecasting model. The theory is explained intuitively and is easy to follow, the results are well-explained and graphically represented. I’d like to express my respect for the effort done by the author with regard to this manuscript. Please see my several critical comments as an effort to further improve the manuscript.

Shortcomings

My major point of criticism is the integration of the present results into the existing literature. Some claims are too strong and overselling the results, moreover in several parts I’d strongly suggest to consider additional relevant literature. Please find more details on this point below

Abstract: I find the claim “the principles governing optimal wagering have not been presented” way too strong. Please be more careful and precise here. Please also find more information on this point below.

p. 1 Introduction l. 5: The author states “important insights into market efficiency”. In my mind this is simplifying as the results of market efficiency papers can point into different directions. So I would suggest to be more careful here by stating that market efficiency has been the subject of investigation or more precise by saying what the important insights were. Moreover, the author states three papers from 1968, 1997 and 2004. Below, I have summarised some more recent papers that might be worth considering:

Angelini, G., & De Angelis, L. (2019). Efficiency of online football betting markets. International Journal of Forecasting, 35(2), 712-721.

Bernardo, G., Ruberti, M., & Verona, R. (2019). Semi-strong inefficiency in the fixed odds betting market: Underestimating the positive impact of head coach replacement in the main European soccer leagues. The Quarterly Review of Economics and Finance, 71, 239-246.

Meier, P. F., Flepp, R., & Franck, E. P. (2021). Are sports betting markets semistrong efficient? Evidence from the COVID-19 pandemic. International Journal of Sport Finance, 16(3).

p. 1 Introduction 2nd paragraph: The author claims that literature on “optimal decision-making from the bettor’s perspective” is missing. From my point of view, this is overselling the novelty of the current manuscript. While I certainly see what the manuscript adds to the literature, I would like to see acknowledged that decisions from a bettor’s perspective have been considered in the literature explicitly and implicitly.

Examples are the well-known Kelly betting strategy for optimal stake sizes going back to

Kelly, J. L. (1956). A new interpretation of information rate. the bell system technical journal, 35(4), 917-926.

or literature focusing on the effects of non-optimal decisions of bettors

Snowberg, E., & Wolfers, J. (2010). Explaining the favorite–long shot bias: Is it risk-love or misperceptions?. Journal of Political Economy, 118(4), 723-746.

Moreover, it is well established in the forecasting literature to use and test several models for deciding on bets such as several stake sizes (e.g. UNIT BET, UNIT WIN, KELLY) in this example:

Hvattum, L. M., & Arntzen, H. (2010). Using ELO ratings for match result prediction in association football. International Journal of forecasting, 26(3), 460-470.

p.3 Theorem 2: I am surprised to not see the following paper cited.

Wunderlich, F., & Memmert, D. (2020). Are betting returns a useful measure of accuracy in (sports) forecasting?. International Journal of Forecasting, 36(2), 713-722.

For example, Theorem 2 is highly related to the area of no profitable bet presented in the aforementioned paper. In general, I see a lot of overlap between the two papers, both analysing betting decisions both from a theoretical point and based on real-world data. The author could also describe how the current manuscript is different from the aforementioned paper, e.g. by using point spreads and by analysing quantiles.

p.8 Bias-variance in sports wagering: I really like the statement that bettors just need their estimation to be on the correct side of the spread, a fact that is often overlooked in profitable forecasting (see also p. 4 last paragraph). You might discuss that the aforementioned paper of Wunderlich & Memmert and the paper of Hubacek & Sir below make similar statements.

Hubáček, O., & Šír, G. (2023). Beating the market with a bad predictive model. International Journal of Forecasting, 39(2), 691-719.

Further points

Title: Why do you use the wording “sports wagering”? As far as I am concerned, this is pretty uncommon in the literature and – unless there is a specific reason that I am not aware of – I would rather expect the wording “sports betting”.

Results Problem formulation point spread: You mention the word point spread betting before giving an explanation on how such bets work. You might want to give a very brief explanation on this before, particularly as a lot of literature in this domain is concentrated on European sports betting markets, where point spreads are not that pronounced.

p.2: Please define or introduce Phi_h, Phi_v is before the first usage.

p. 4 Optimal estimation of the margin of victory: It is assumed that the difference between home and away team m and its estimation are independent. I am neither convinced that this is true nor convinced that this is false. But I am a bit sceptical as this is obviously a strong assumption needed for the further proof. Could you discuss this issue and explain in more detail why this assumption is reasonable?

p.5 Optimality in moneyline wagering: At this point, again, I would suggest to add a reference to European betting markets, where bets without spread (i.e. s=0) are the most common bets. This is also reflected in the literature, which (for example in soccer) is highly concentrated on home, draw, away betting. I would also suggest to state possible differences between (European) home, draw, away and (North American) moneyline betting.

Koopman, S. J., & Lit, R. (2019). Forecasting football match results in national league competitions using score-driven time series models. International Journal of Forecasting, 35(2), 797-809.

Hvattum, L. M., & Arntzen, H. (2010). Using ELO ratings for match result prediction in association football. International Journal of forecasting, 26(3), 460-470.

Constantinou, A. C., & Fenton, N. E. (2013). Determining the level of ability of football teams by dynamic ratings based on the relative discrepancies in scores between adversaries. Journal of Quantitative Analysis in Sports, 9(1), 37-50.

…among many many others

p. 6 Empirical results: I would strongly suggest to state the source of data where you obtained information on > 5.000 matches. Is it data provided by a company or data openly available online? On p. 9 Materials and Methods it appears that the source is the company usually not opening up on the data. However, I would like to see this explained more clearly.

Same paragraph: This paragraph states means and medians from the data. While the manuscript generally correctly underlines the potential difference between mean and median (e.g. due to strongly skewed distributions), the numbers seem to suggest that the real-world distributions are only very weakly skewed and as such mean and median are closely related (e.g. mean 2.19, median 3; mean 44.43, median 44). I would highly like to see this aspect explained and acknowledged.

Same paragraph: The paragraph says that “The standard deviation […] is nearly 7x the mean, indicating the frequent occurrence of outliers (“blowouts”). This claim, in my mind, is at least misleading. While I agree that a high standard deviation indicated frequent blowouts, this has nothing to do with the mean margin of victory which is rather an estimate of home advantage. If the data would show 0.12 + 14.68 instead of 2.19 + 14.68, would this be a sign of even more blowouts???

p. 7 last paragraph: I really like that the author states that (besides correct forecasting) the bookmaker might have other incentives such as risk management (i.e. book balancing). I wonder and I would like to see discussed whether there might be additional incentives of the bookmaker that contradict perfect forecasting. Just as one example, bookmakers might chose higher odds than reasonable for marketing reasons in some specific games. Might bookmakers in spread betting favour completely equal odds over slightly different odds, although not representing their true belief in the probabilities? To be very precise here, this is a true question from my side, i.e. I don’t want to express that this is actually the case.

p. 8 The case for quantile regression: The manuscript states “substantial deviation between the mean and median“. This seems to be in contradiction to the data presented in the results section (see three points before). Please adjust this statement or give a better explanation on why you think this is the case.

General point: Not related to any specific part of the manuscript, but you might want to discuss differences between American football and other sports. In terms of forecasting and statistical modelling American football is a very specific challenge as it has different possibilities to score (field goal, touchdown, extra point etc.) while other sports usually have only one possibility to score.

7. PLOS authors have the option to publish the peer review history of their article (what does this mean?). If published, this will include your full peer review and any attached files.

Reviewer #2: No

Reviewer #3: **Yes: **Dr. Fabian Wunderlich

---

## [Editor Report · Decision Letter 2]

8 Jun 2023

A statistical theory of optimal decision-making in sports betting

PONE-D-22-34727R2

Dear Dr. Dmochowski,

We’re pleased to inform you that your manuscript has been judged scientifically suitable for publication and will be formally accepted for publication once it meets all outstanding technical requirements.

Kind regards,

Baogui Xin, Ph.D.

Academic Editor

PLOS ONE
---

## [Editor Report · Acceptance letter]

19 Jun 2023

PONE-D-22-34727R2 

A statistical theory of optimal decision-making in sports betting 

Dear Dr. Dmochowski:

I'm pleased to inform you that your manuscript has been deemed suitable for publication in PLOS ONE. Congratulations! Your manuscript is now with our production department. 

Kind regards, 

on behalf of

Professor Baogui Xin 

Academic Editor

PLOS ONE